

# Estimating the impact of mobility patterns on COVID-19 infection rates in 11 European countries

Patrick Bryant[1,2] and Arne Elofsson[1,2]

[1] Department of Biochemistry and Biophysics, Stockholm University, Stockholm, Sweden
[2] Science for Life Laboratory, Solna, Sweden

## ABSTRACT

**Background:** As governments across Europe have issued non-pharmaceutical interventions (NPIs) such as social distancing and school closing, the mobility patterns in these countries have changed. Most states have implemented similar NPIs at similar time points. However, it is likely different countries and populations respond differently to the NPIs and that these differences cause mobility patterns and thereby the epidemic development to change.

**Methods:** We build a Bayesian model that estimates the number of deaths on a given day dependent on changes in the basic reproductive number, $R_0$, due to differences in mobility patterns. We utilise mobility data from Google mobility reports using five different categories: retail and recreation, grocery and pharmacy, transit stations, workplace and residential. The importance of each mobility category for predicting changes in $R_0$ is estimated through the model.

**Findings:** The changes in mobility have a considerable overlap with the introduction of governmental NPIs, highlighting the importance of government action for population behavioural change. The shift in mobility in all categories shows high correlations with the death rates 1 month later. Reduction of movement within the grocery and pharmacy sector is estimated to account for most of the decrease in $R_0$.

**Interpretation:** Our model predicts 3-week epidemic forecasts, using real-time observations of changes in mobility patterns, which can provide governments with direct feedback on the effects of their NPIs. The model predicts the changes in a majority of the countries accurately but overestimates the impact of NPIs in Sweden and Denmark and underestimates them in France and Belgium. We also note that the exponential nature of all epidemiological models based on the basic reproductive number, $R_0$ cause small errors to have extensive effects on the predicted outcome.

## INTRODUCTION

In December 2019 a new coronavirus (COVID-19) emerged in Wuhan, China. China implemented a quick strategy of suppression by imposing a lockdown in the city of Wuhan on January 23 (https://www.reuters.com/article/us-china-health-who-idUSKBN1ZM1G9, last accessed 1 May 2020), and implementing social distancing procedures nationwide,

Corresponding author
Patrick Bryant,
patrick.bryant@scilifelab.se

with a successful outcome (*Li et al., 2020*). Still, the virus rapidly spread across the world through our increasingly interconnected flight network, and shortly arrived in Europe. In February 2020 the number of cases started to increase quickly in some European countries. European countries introduced non-pharmaceutical interventions (NPIs) similar to those used in China to limit the spread of the virus. These NPIs include social distancing, school closures, restrict international travel and lockdown (*European Centre of Disease Control, 2020*). The NPIs results in behavioural changes, and these can be traced by tracking the location of mobile phones.

After an initial rapid spread in China, control measures proved very successful to stop the spread both in China (*Lai et al., 2020*) and in other parts of the world (*Milne & Xie, 2020*; *Flaxman et al., 2020a*). However, there is still a risk for subsequent infections upon lifting of these restrictions (*Flaxman et al., 2020a*; *Ferguson et al., 2020*). There is, therefore, an urgent need both for understanding and tracking the effects of governmental interventions and their removals. Largescale testing could provide valuable information about the impact of interventions. However, these are expensive, sometimes inaccurate and might violate privacy rights. In contrast, the use of largescale data from anonymous tracking of mobile phones is inexpensive and readily available.

Google recently released a time-limited sharing of mobility data (https://www.google.com/covid19/mobility, last accessed 29 March 2020) from across the world as represented by summary statistics to combat COVID-19. The mobility data is measured in six different sectors: retail and recreation, grocery and pharmacy, parks, transit stations, workplace and residential. The effects of the government-issued NPIs can be seen through changes in these patterns.

It is likely that different countries respond in different manners to the same NPIs, why it is vital to consider the effect of NPIs country wise. Here, we show that by using real-life mobility data to model changes in the basic reproductive number, $R_0$, the impact of NPIs across different countries can be modelled more accurately. The mobility data utilised here have some uncertainties and lack resolution. Still, to the best of our knowledge, this data is the best openly available data source for tracking a population's movement in the 11 studied countries. Governments can, in collaboration with telephone companies, obtain much more fine-grained data, enabling them to evaluate the effect of the NPIs in more detail.

Recently, a group from Imperial College released a report (*Flaxman et al., 2020a*) that estimates the effects of NPIs on $R_0$. Subsequently, a modified version of this report was published (*Flaxman et al., 2020b*). The report had a massive impact on how the UK government changed its intervention strategy (https://www.imperial.ac.uk/news/196477/j-ideas-neil-ferguson-tells-mps-lockdown/, last accessed 1 May 2020). A limitation of the Imperial College London (ICL) model is the assumption that each intervention has the same impact in all countries, ignoring cultural and sociological differences as well as differences in the details of the NPIs. Here, we try to overcome this by developing an extension to their model utilising country-specific mobility data in a Bayesian framework (*Banerjee, Carlin & Gelfand, 2015*), we estimate the impact of each change in mobility pattern on $R_0$. The resulting information provides a smooth, straightforward way for

governments to analyse if NPIs are working and to what extent. We show that in a 3-week forecast, our method makes a better prediction than the model from Imperial College.

## METHODS

Here, we introduce an Markov-Chain Monte-Carlo (MCMC) model to estimate the spread of the COVID-19 infection in various countries. The ICL model strongly inspires the model, and all parameters are taken from earlier studies. For each country, we define a starting point when the total number of observed deaths has reached 10. The model is trained using data starting 30 days before this day and until 29 of March 2020. Finally, the model is used to simulate a 3-week forecast from 30 March to 19 April.

### Infection model

The number of cases acquired at day $\tau$ in country $m$, $c_{\tau,m}$ is modelled with a discrete renewal process (*Fraser, 2007*; *Cauchemez et al., 2008*):

$$c_{\tau,m} = R_{\tau,m} \sum_{\tau=0}^{t-1} c_{\tau,m} g_{\tau-t} \tag{1}$$

where

$$g_{\tau-t} \sim \text{Gamma}(6.5, 0.62) \tag{2}$$

(Gamma distribution with a mean of 6.5 days and a standard deviation of 0.62 days) is the serial interval distribution used to model the number of cases (*Flaxman et al., 2020a*; *Backer, Klinkenberg & Wallinga, 2020*).

$g_s$ is discretized in steps of 1 day accordingly:

$$g_s = \int_{\tau=s-0.5}^{s+0.5} g(\tau)d\tau \; for \; s \; = \; 2, 3, \ldots \; and \; g_1 = \int_{\tau=0}^{1.5} g(\tau)d\tau \tag{3}$$

The number of cases today is thus dependent on the cumulative number of cases from the previous days, weighted by the serial interval distribution, multiplied with the basic reproductive number ($R_0$) at day $t$. The discretizations, here and elsewhere, of 1 day are motivated by the intervals in reporting. Just as in the ICL model (*Flaxman et al., 2020a*), we assume the starting point for the infection was 30 days before the day after each country has observed 10 deaths in total. The time delay of 30 days is necessary due to the relationship between infection and death (see Death model described below). From this assumed starting point, we initialise our model with 6 days (*Li et al., 2020*) of cases drawn from an Exponential (0.03) distribution, which are inferred in the Bayesian posterior distribution ($D_{t,m}$).

### Impact on the basic reproductive number

Our model is based on the model used in the recent report (*Flaxman et al., 2020a*) from ICL. The ICL report tries to estimate the impact of NPIs on $R_0$ in the same 11 countries modelled here. The main difference between the ICL model and the current one is the modelling of the change of $R_0$. In the ICL model, the basic reproductive number at day $t$ in

country $m$ ($R_{t,m}$) is estimated as a function of the NPI indicators $I_{k,t,m}$ in place at day $t$ in country $m$ as:

$$R_{t,m} = R_{0,m} e^{-\sum_{I=1}^{6} \alpha_k I_{k,t,m}} \qquad (4)$$

where $I = 1$ when intervention $k$ is implemented at day $t$ in country $m$ and $\alpha$ the impact of each intervention.

Here, we instead estimate $R_{t,m}$ to be a function of the relative change in mobility pattern for each country:

$$R_{t,m} = R_{0,m} e^{\alpha_1 I_{1,t,m} + \alpha_2 I_{2,t,m} + \alpha_3 I_{3,t,m} + \alpha_4 I_{4,t,m} - \alpha_5 I_{5,t,m}} \qquad (5)$$

where $I_{1-5,t,m}$ is the relative mobility in retail and recreation, grocery and pharmacy, transit stations, workplace and residential sectors respectively at day $t$ in country $m$. The residential mobility parameter has a negative sign as it is assumed that when people stay at home it lowers $R_0$. In our model, we assume that the impact of each relative mobility change has the same relative impact across all countries and across time. This assumption is a requirement to enable the estimation of the impact of mobility on $R_0$. If the mobility impacts were allowed to differ between countries, it would not be possible to discern between other country-specific factors and the effect of changes in mobility.

The parameter alpha is set to be gamma distributed with mean 0.5 and a standard deviation of one. A narrow gamma distribution was chosen due to the assumption that the impact on $R_0$ is almost instantaneous, with an effect that decreases quickly with time. We did not include the data for the mobility category 'Parks' as this data displayed much noise and cyclic peaks, possibly caused by varying weather (https://www.google.com/covid19/mobility, last accessed 29 March). The prior for $R_0$ is set to:

$$R_{0,m} \sim Normal(2.79 | \kappa), \ with \ \kappa \sim Normal(0, 0.5) \qquad (6)$$

The value of 2.79 is chosen from the median value of a recent analysis of 12 modelling studies (*Liu et al., 2020*), and the normal distribution from (*Li et al., 2020*).

The relative mobility is modelled as the relative value change compared to a mobility baseline estimated by Google (https://www.google.com/covid19/mobility, last accessed 29 March). The baseline is the median value, for the corresponding day of the week, during the 5-week period of 2020-01-03 to 2020-02-06. For the days for which no mobility data is available, the values were set to zero. The mobility data for the forecast (and days beyond the date for the last available mobility data) was set to the same values as the last observed days. The time points for the interventions were taken from the ICL report (*Flaxman et al., 2020a*), whose initial efforts were crowdsourced.

## Death model

As the number of deaths in each country is likely to be the most accurate COVID-19 related data, we use this as the core of the model, being the posterior in the Bayesian simulations. The number of deaths in country $m$ at day $t$ is modelled as a negative binomial

distribution as used in earlier models (*Fraser, 2007*; *Lloyd-Smith et al., 2005*) with mean and variance accordingly:

$$D_{t,m} \sim Negative\ Binomial\left(d_{t,m}, \frac{d_{t,m}^2}{\psi}\right), \ \psi \sim Normal^+(0,5) \tag{7}$$

The expected number of deaths, $d_{t,m}$, at day $t$ in country $m$ is given by:

$$d_{t,m} = \sum_{\tau=0}^{t-1} c_{\tau,m}\pi_{t-\tau,m} \tag{8}$$

where $\pi_m$ is the infection to death distribution in the country $m$ given by a combination of the infection to onset distribution (Gamma(5.1, 0.86)) and onset to death distribution (Gamma(17.8, 0.45)) (combined with mean 22.9 days and standard deviation 0.45 days) times the infection fatality rate (*ifr*) (*Flaxman et al., 2020a*; *Verity et al., 2020*; *Lauer et al., 2020*):

$$\pi_{t,m} \sim ifr_m \cdot Gamma(5.1 + 17.8, \ 0.45) \tag{9}$$

$\pi_{t,m}$ is discretized in steps of 1 day accordingly:

$$\pi_{s,m} = \int_{\tau=s-0.5}^{s+0.5} \pi_m(\tau)d\tau \ for \ s \ = \ 2, 3, \dots \ and \ \pi_{1,m} = \int_{\tau=0}^{1.5} \pi_m(\tau)d\tau \tag{10}$$

The *ifrs* are taken from previous estimates of the population at risk is about 1% (*Lourenco et al., 2020*) and adjusted for the predicted attack rate in the age group 50–59 years of age, assuming a uniform attack rate (*Flaxman et al., 2020a*; *Ferguson et al., 2020*; *Verity et al., 2020*), chosen due to having the least predicted underreporting in analyses of data from the Chinese epidemic (*Verity et al., 2020*). The uniform attack rate is required due to a lack of age-specific data. The number of deaths today is thus dependent on the cumulative number of cases from the previous days, weighted by the country-specific infection to death distribution.

The implications on $R_0$ due to relative mobility variations were estimated simultaneously for all countries in a hierarchical Bayesian framework using MCMC (*Banerjee, Carlin & Gelfand, 2015*) simulations in Stan (*Stan, 2020*). The death data (https://www.ecdc.europa.eu/en/publications-data/download-todays-data-geographic-distribution-covid-19-cases-worldwide, last accessed 19 April 2020) used in the form of the number of deaths per day is from European Centre of Disease Control (ECDC), available and updated daily. We ran the model with eight chains, using 4,000 iterations (2,000 warm-up), as in the earlier work (*Flaxman et al., 2020a*; *Stan, 2020*). The parameter specifics of the simulation are available in the code (see below).

## MCMC convergence

Markov-Chain Monte-Carlo (MCMC) simulations are considered to converge when the Rhat statistics (a metric for comparing the variance between pooled and within-chain inferences) reach one (*Brooks & Gelman, 1998*). A histogram of Rhat statistics for the

modelled parameters in all simulation runs were constructed and analysed. We also ensure that no divergent transitions were observed by setting the adapt delta in the sampler (see code).

### Leave one country out analysis

Since all countries are in different stages of their epidemics, different amounts of data are available for each country. To analyse how the model is influenced by different countries, we fit models using data from all countries except one using all 11 combinations (*Hastie, Tibshirani & Friedman, 2013*). We then estimate the importance of each mobility parameter in the leave-one-country-out (LOO) analysis. The relative difference in each mobility parameter provides an estimate of how each country affects $R_0$ and thus the number of cases and deaths as well. Furthermore, the Pearson correlation coefficients for the mean $R_0$ across all time points are calculated for each country in the different runs when the other 10 were left out (see Fig. S1).

### Forecast validation

To ensure the forecasts are reliable, we leave out 3 weeks of data (30 March–19 April) and fit a model using data from the beginning of the epidemic up to the date for the beginning of the left-out data. We then evaluate the model with 1-week intervals from the 30th of March to the 19th of April. We evaluate by the average error and the average fractional error (average error ÷ Σobserved deaths) during each of the 3 weeks. We compare our results with simulations obtained from the ICL model (*Flaxman et al., 2020a*). We should note here that the ICL model does not converge for 3-week predictions using 4,000 iterations (see Fig. S2).

### EpiEstim estimates of the basic reproductive number ($R_0$)

To validate our estimates of $R_0$, we estimate $R_0$ independently using case data from ECDC and the R package EpiEstim (*Thompson et al., 2019*), based on the SIR model (*McKendrick, 1925*). The serial interval used for the estimations is variable accordingly: estimate_R (country_cases, method="uncertain_si", config = make_config(list(mean_si = 7.5, std_mean_si = 2, min_mean_si = 1, max_mean_si = 8.4, std_si = 3.4, std_std_si = 1, min_std_si = 0.5, max_std_si = 4, n1 = 1,000, n2 = 1,000))), allowing more possible scenarios to be explored (see code, "Methods" section). The $R_0$ estimates are smoothed using 1-week averages, since they are uncertain in the beginning of the epidemic when cases are few. These values are compared with those of the mobility model, only including values under five due to the high uncertainty of the larger values in the beginning of the epidemic. The correlations are high and the average errors are low, mainly arising in areas of large uncertainties (see Fig. S8).

### Correlation analysis

To ensure that there is a true relationship between the daily deaths and the mobility changes, correlations between the deaths per day and the different mobility parameters were analysed. Both the death data and the mobility data were first smoothed using 1-week averages. The correlations were made by shifting the daily deaths to infer the time delay of
which type of mobility affects the daily deaths. The shifts are from 0 to 48 days, ensuring all countries have at least 10 days of data for the correlation analysis. The correlations, without shifts, between the different mobility parameters, were also analysed (see Fig. S3).

### Code

The code is written in Python using the Stan package pystan (v. 2.19.1.1) for MCMC simulations. The code is freely available under the GPLv3 licence. https://github.com/patrickbryant1/COVID19.github.io/tree/master/simulations/mobility.

## RESULTS

### Estimating the cumulative number of cases, the number of deaths per day and changes in the basic reproductive number, $R_0$

In Fig. 1, for Italy and Sweden, and Fig. S4, for all 11 modelled countries, estimates of cumulative cases, daily deaths and the basic reproductive number $R_0$ are shown.
We simulate a 3-week forecast from 30 March to 19 April using data up to 29 March from the ECDC in the form of the number of deaths per day, and relative mobility data estimated by Google (https://www.google.com/covid19/mobility, last accessed 29 March). According to the model, most countries appear to have their epidemic under control (April 19) (Table 1). The most successful nation in terms of reducing $R_0$ is Italy ($R_0 \approx 0.22$), and the least is Sweden ($R_0 \approx 2.01$).

From Fig. S4, it can be seen that in all countries, the interventions have some positive effect, decreasing the estimated $R_0$ between the epidemic start and March 29. It can be noted that during the development of the epidemic, $R_0$ displays a wide range of values. In some countries, the mean of the estimated $R_0$ shows a rapid increase to values as high as 10, coupled with an increase in mobility (primarily) to grocery and pharmacies exactly when the interventions were introduced. Most posterior distributions for the mean $R_0$ values are centred around the prior of 2.79 (Fig. 2). Notable is that Italy and Spain, which both had very rapid spread have distributions centred higher than the prior.

The estimated number of deaths for up to 3 weeks after the model is trained, have a good correspondence with the observed number (Fig. 1; Fig. S4; Table 2). Compared with the ICL model (*Flaxman et al., 2020a*), our model displays both lower errors and less uncertainty (see Fig. 3; Fig. S5; Table S1). The average absolute errors over the 11 countries in the number of deaths are lower across all 3 weeks (week 1: 60 vs 159, week 2: 95 vs 472, and week 3: 88 vs 1,429 for our model and the ICL model respectively).

### Comparing mobility data across countries

When overlaying the implementation dates of the NPIs with the mobility data, it is clear that governmental decisions have a significant impact on the populations in the 11 modelled countries (see Fig. S4). Most countries display very similar relative changes in their mobility patterns, with mobility in retail and recreation, grocery and pharmacy, transit stations and workplace decreasing, while mobility in the residential category is increasing.

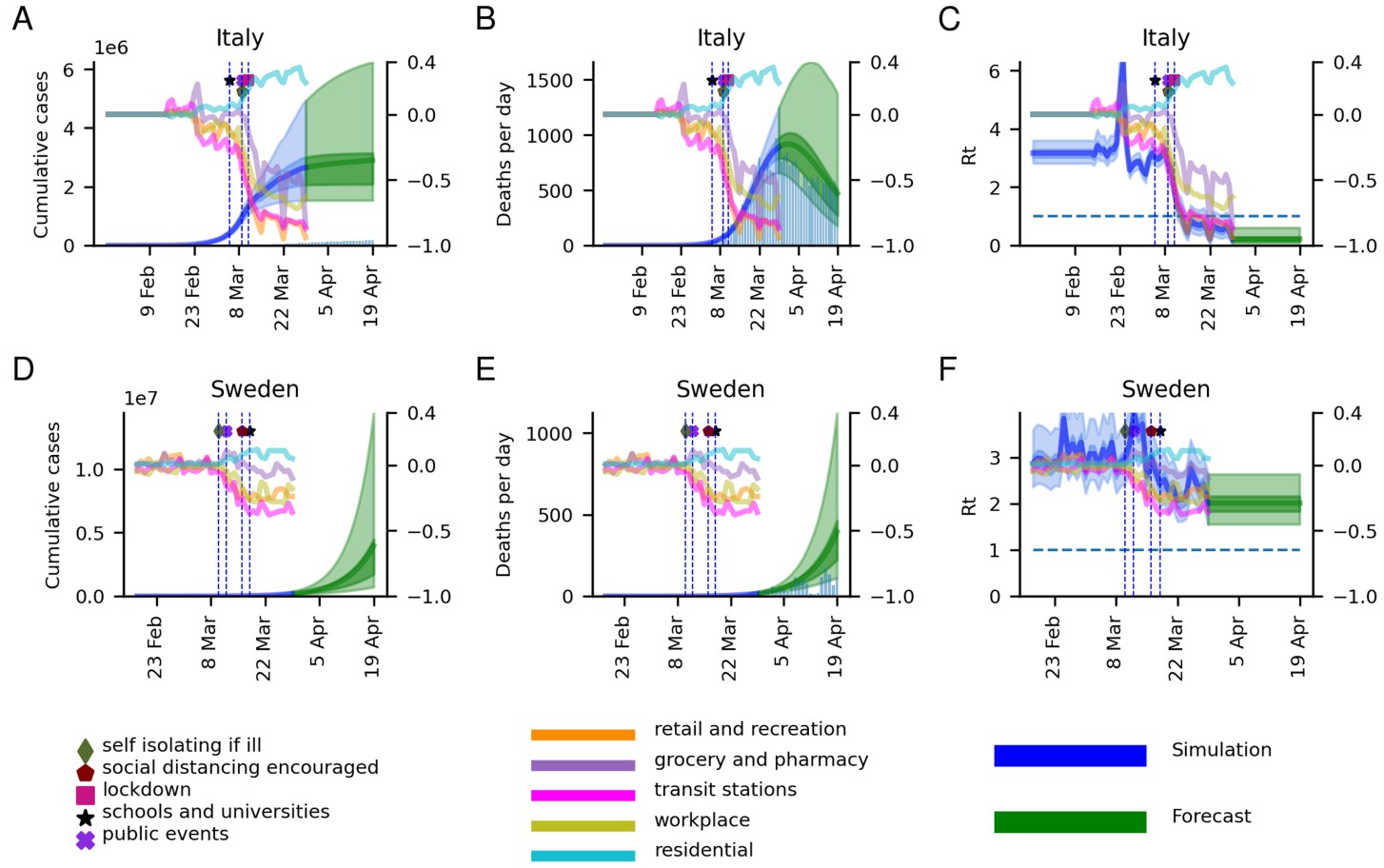

**Figure 1 Model results for Italy and Sweden.** Model results in the form of the cumulative number of cases, deaths per day and $R_0$ for Italy (A–C) and Sweden (D–F), are displayed on the left axes. The model results start from 30 days before 10 accumulated deaths had been observed. The blue curves represent the estimations so far, while the green represents a 3-week forecast (30 March–19 April). The 50% and 95% confidence intervals are displayed in darker and lighter shades respectively, with the mean as a solid line. The histograms represent the number of cases and deaths reported by the European Center for Disease Control (ECDC). Mobility data for the five modelled sectors represented in terms of relative change compared to baseline (observed in a 5-week period of 2020-01-03 to 2020-02-06) is displayed on the right axes. The dates for the introduction of different NPIs are marked with vertical lines. As can be seen, the NPIs have very strong implications for the mobility patterns. The mobility data ranges from 2020-02-15 to 2020-03-29, after which the final levels are fixed. The graph for Rt includes a dashed horizontal line marking the value one of halted epidemic growth.

Most countries have similar relative changes across the sectors (Fig. S4). The ones that display smaller relative changes (Denmark, Norway and Sweden) also demonstrate more modest reductions in $R_0$, which is a natural consequence of our model, as it assumes that changes in $R_0$ are directly related to changes in mobility. The mobility patterns in Sweden display barely half of the relative changes compared with France, Spain, and Italy, and the reduction in $R_0$ is, therefore, smaller in Sweden.

**The importance of mobility sectors for modelling changes in $R_0$**

Analyzing the importance of each mobility parameter for predicting the reduction in $R_0$ $(1 - e^{-\text{alpha}})$ shows that the grocery and pharmacy sector appears to be the clearest indicator for $R_0$ change (see Fig. 4). The grocery and pharmacy sector is estimated to account for most of the reduc revision2_trackedtion of $R_0$, with a median reduction of

**Table 1 Changes in $R_0$ and mobility in the grocery and pharmacy sector during the epidemic.**

| Country | Modelled start of the epidemic | Estimated mean $R_0$ at epidemic start | Estimated mean $R_0$ at 29 March | Relative change in groceries and pharmacies on 29 March (%) |
| --- | --- | --- | --- | --- |
| Austria | 2020-02-22 | 3.11 | 0.36 | −64 |
| Belgium | 2020-02-18 | 3.24 | 0.51 | −53 |
| Denmark | 2020-02-21 | 3.02 | 1.36 | −22 |
| France | 2020-02-07 | 2.91 | 0.30 | −72 |
| Germany | 2020-02-15 | 3.08 | 0.56 | −51 |
| Italy | 2020-01-27 | 3.17 | 0.22 | −85 |
| Norway | 2020-02-24 | 2.82 | 0.92 | −32 |
| Spain | 2020-02-09 | 3.19 | 0.29 | −76 |
| Sweden | 2020-02-18 | 2.89 | 2.01 | −10 |
| Switzerland | 2020-02-14 | 2.81 | 0.53 | −51 |
| United Kingdom | 2020-02-12 | 2.82 | 0.61 | −46 |

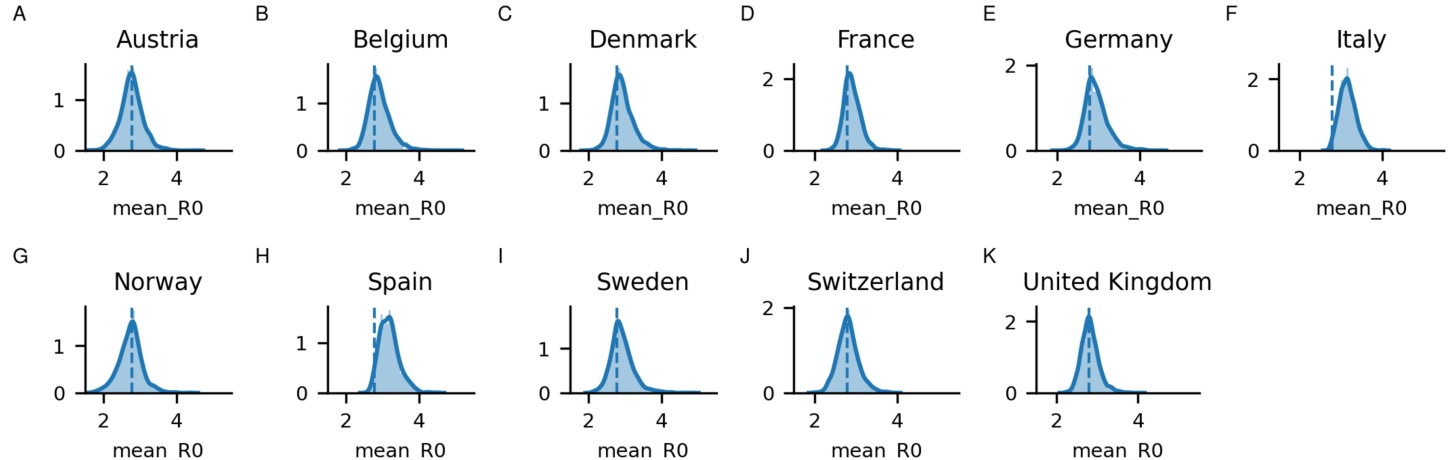

**Figure 2 Posterior distributions for the mean initial $R_0$ sampled per country (A) Austria, (B) Belgium, (C) Denmark, (D) France, (E) Germany, (F) Italy, (G) Norway, (H) Spain, (I) Sweden, (J) Switzerland, and (K) United Kingdom.** The dashed line corresponds to the prior mean, set to 2.79.

95.6% compared to less than 10% for the other sectors (retail and recreation 3.8%, transit stations 3.0%, workplace 4.0%, residential 7.9%).

Investigating the correlation between the deaths per day and the different mobility parameters (Fig. 5), one can see that all sectors display high opposite correlations with a shift of about 20 days. These correlations are due to the time-delayed relationship between the initial spread of the disease, causing deaths occurring after the reduction in mobility, see Fig. S4. The mobility changes have the highest correlations with the number of deaths 30–40 days after they occur, suggesting that the mobility affects the death rate with a time delay of 30–40 days. Roughly in agreement with the 22.9 days in our model. Since the grocery and pharmacy sector displays the most significant correlations,

**Table 2 Average error and average fractional error in the number of deaths for the mobility model.**
Average error and average fractional error in the number of deaths for each country between the mean predicted number of deaths per day and the observed number in 1, 2 and 3 week forecasts, respectively. A corresponding table for the ICL model can be found in Table S2.

**Three-week predictions for the number of deaths per day**

| Country | Average error | | | Average fractional error | | |
|---|---|---|---|---|---|---|
| | Week 1 | Week 2 | Week 3 | Week 1 (%) | Week 2 (%) | Week 3 (%) |
| Austria | −3 | −6 | −2 | −2.3 | −4.0 | −1.7 |
| Belgium | −46 | −179 | −186 | −5.0 | −8.7 | −8.8 |
| Denmark | 0 | 10 | 28 | 0.4 | 10.2 | 32.2 |
| France | −318 | −445 | −427 | −6.1 | −7.1 | −7.8 |
| Germany | −21 | −7 | −26 | −2.2 | −0.5 | −1.6 |
| Italy | 144 | 201 | 29 | 2.7 | 4.9 | 0.8 |
| Norway | 1 | 1 | 3 | 1.7 | 2.2 | 6.9 |
| Spain | −98 | 84 | −8 | −1.6 | 1.8 | −0.2 |
| Sweden | −3 | 28 | 180 | −1.2 | 5.4 | 28.9 |
| Switzerland | 13 | 41 | 48 | 4.3 | 14.1 | 17.1 |
| United Kingdom | 17 | −42 | 32 | 0.4 | −0.7 | 0.5 |
| Average absolute error | 60 | 95 | 88 | 2.5 | 5.4 | 9.7 |

the model assigns most weight to that sector, although the mobility in all sectors is highly correlated with each other (Fig. S3).

## Model validation

The posterior distributions for the mobility parameters (see Fig. S6) are almost identical in the LOO analysis. A bimodal distribution is observed when leaving Italy out in the grocery and pharmacy sector though, emphasising the importance of the Italian data. The variable $R_0$ values in the LOO analysis show Pearson correlations close to one, with Italy and especially the United Kingdom displaying lower correlations of around 0.8 and consistently below 0.8 respectively (see Fig. S1). Italy and the United Kingdom correlate badly with each other, with Pearson correlations of close to 0. 11 of 4,000 iterations ended with a divergence (0.275%) Spain was excluded. A histogram of Rhat statistics for the modelled parameters in all simulations for the main analysis is displayed in Fig. S7.

To validate the $R_0$ estimates, we used a SIR model using EpiEstim (*Thompson et al., 2019*) to estimate $R_0$ independently from case data (and not death data as in our and the ICL models). This model does not try to determine the cause of changes in $R_0$, but just estimates the changes from the number of reported cases. In general, the overlap of the two estimates of $R_0$ estimates is high, in particular at the crucial time points before and after the effects of NPI implementation (see Table S2; Fig. S8). Denmark, Norway and Spain display the most substantial differences between the estimates, differing 2.98, 1.94 and 3.48 respectively at the point before NPI implementation. The differences that do

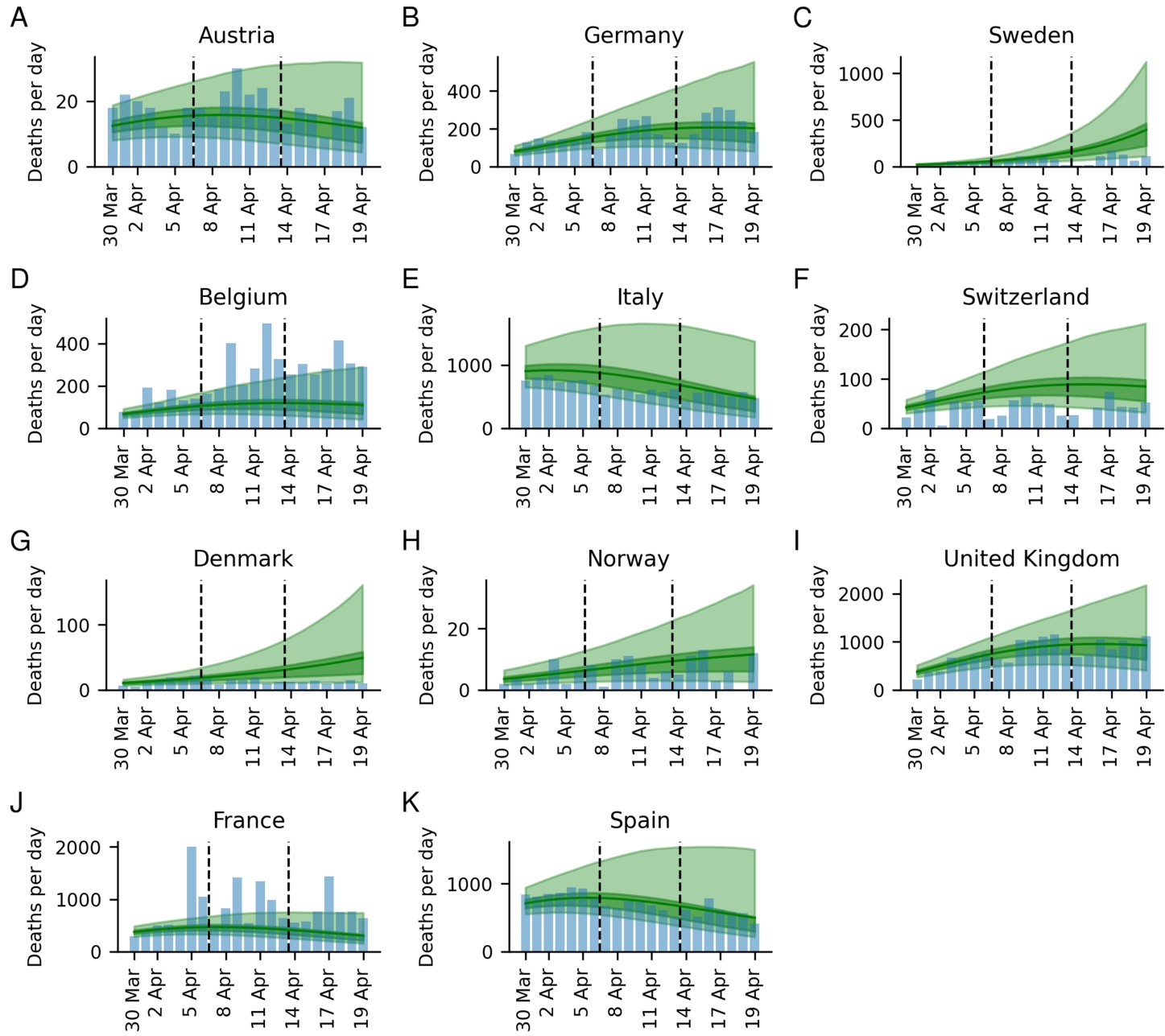

**Figure 3 Three-week predictions for all countries.** Three-week predictions for all countries, (A) Austria, (B) Germany, (C) Sweden, (D) Belgium, (E) Italy, (F), Switzerland, (G) Denmark, (H) Norway, (I) United Kingdom, (J) France, and (K) Spain, in the form of deaths per day for the weeks 1: (Mar 30–April 5), week 2 (April 6–April 12) and week 3 (April 13–April 19)/. The 50% and 95% confidence intervals are displayed in darker and lighter shades respectively, with the mean as a solid line. The blue histogram represents the observed deaths.

arise are mainly during the periods with considerable uncertainty in the $R_0$ estimates, that is when the number of reported cases is low. Sweden shows the most substantial error between the estimates after NPI implementation (0.98). Further, the models show very different speeds of the changes in $R_0$ values, EpiEstim having a much slower response than the mobility model.

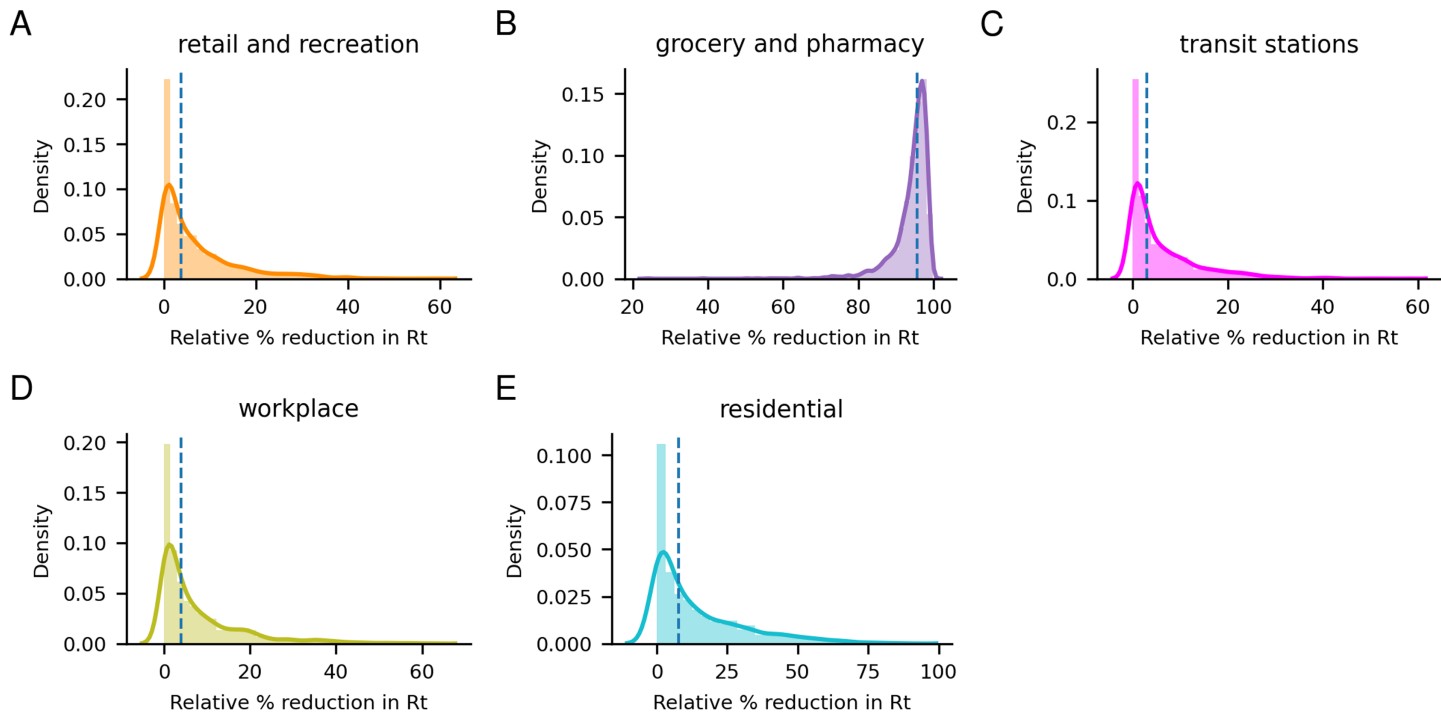

**Figure 4** **Posterior distributions of the impact of each mobility parameter.** Posterior distributions of the impact of each mobility parameter for predicting the reduction in $R_0$. The grocery and pharmacy sector appears to be the clearest indicator for $R_0$ change. The median impacts are 3.8%, 95.6%, 3.0%, 4.0% and 7.9% for the (A) retail and recreation, (B) grocery and pharmacy, (C) transit, (D) workplace and (E) residential sectors respectively.                                                                     

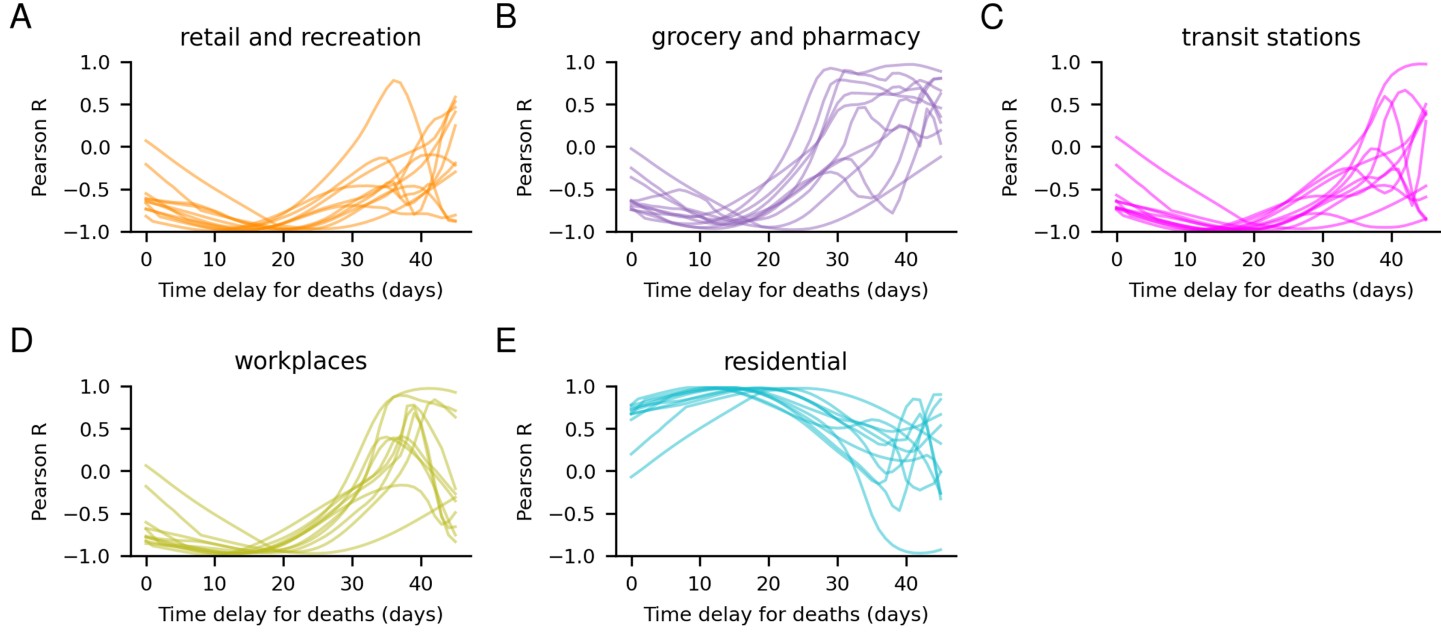

**Figure 5** **Correlation between daily deaths and mobility changes in different sectors (A–E).** Correlation between deaths per day and mobility changes for different time delays. Each country is represented by one line. The mobility changes have the highest correlations with the deaths about 30–40 days after they occur, suggesting that mobility affects the death rate with a time delay of 30–40 days.

## DISCUSSION

The model makes it clear that the NPIs introduced by governments across Europe have had substantial effects on both mobility patterns and in preventing the spread of COVID-19. By tracking the relative change in mobility in the grocery and pharmacy sector, it is possible to account for most of the reduction in the basic reproductive number, $R_0$, in our model. This information can, therefore, provide a useful, straightforward way for governments to analyse the effect of their NPIs.

Why the grocery and pharmacy sector has been assigned the highest importance is likely because this sector displays the strongest correlation with the daily deaths. The correlations are highest assuming a 30–40 day shift, suggesting that mobility affects the death rate with a time delay of 30–40 days, in rough agreement with our model. Since $R_0$ is strongly dependent on the changes in mobility, rapid changes in mobility lead to rapid changes in $R_0$, with drastic consequences to the estimated development of the epidemic in a country. However, changes in $R_0$ will not manifest in the number of deaths per day until about 3 weeks later (the mean value in the gamma distribution for infection to death is 22.9 days, see "Methods" section). Therefore, a 3-week forecast is provided.

The estimates have an acceptable correspondence with the observed numbers in most countries (see Fig. 3; Table 2), and compared with the ICL-model, our model displays both lower errors and less uncertainty (Fig. 3; Fig. S5; Tables 2; Table S1). It can also be noted that the ICL model overpredicts the number of deaths in all countries. The higher accuracy when including mobility data, further suggests the usefulness of our model.

The estimated number of cases has considerable uncertainty across all countries. One limitation of our model is that it does not take herd-immunity effects into account, which should be reached when around 60–80% of the population is infected (*Kwok et al., 2020*). Still, it is unlikely that sufficiently high infection has been reached yet for this to have a significant effect. Another limitation of the model is the assumption that the impact of each relative mobility change has the same relative impact across all countries and across time. If the mobility impact were allowed to differ between countries and in time, it would not be possible to discern between other country-specific and time factors and the mobility impact. Likely both more detailed mobility data and intermixing patterns need to be considered, metrics that are not available.

The number of cases is also highly dependent on having the correct *ifr*. This quantity is only modelled for the age group 50–59 years and does thereby not consider the attack rates for the whole of each country's population (see "Methods" section). If a nation managed to avoid the elderly being infected, that would lower the *ifr* (*Ruan et al., 2020*), which could explain prediction differences to some extent.

The model validation, by a LOO analysis, comparing with independent $R_0$ estimates from EpiEstim (*Thompson et al., 2019*) and predicting a 3-week forecast ensures the model's robustness. The LOO analysis shows that the estimates are mostly affected by the data from Italy and the UK, likely due to these countries having more available data and higher death tolls early in the epidemic, making the model somewhat biased to these data in the beginning of the estimates (Fig. S4). The comparison with the $R_0$ estimates

from EpiEstim show differences that arise mainly during the periods with considerable uncertainty in the $R_0$ estimates, that is when the number of reported cases are low. The estimates also show very different speeds of the changes in $R_0$ values, EpiEstim having a much slower response than the mobility model (Fig. S8).

The countries in the 3-week forecast where the errors stand out are Denmark and Sweden, with over-predictions, and Belgium and France, which are under-predicted. We note that these two pairs of countries are close both geographically and culturally (*Warner-Søderholm, 2012*; *Hofstede & Hofstede, 2001*), possibly explaining the systematic differences. The differences may also be caused by differences in reporting between the countries (https://www.bloomberg.com/news/articles/2020-04-09/french-virus-deaths-jump-with-more-nursing-home-patients-counted, last accessed May 1; https://www.politico.com/news/2020/04/19/why-is-belgiums-death-toll-so-high-195778, last accessed May 1). For instance, on April 5 more than 2,000 deaths were reported in France, due to sudden inclusion of potential COVID-19 attributed deaths in nursing homes occurring at earlier dates (https://www.usnews.com/news/world/articles/2020-04-02/frances-coronavirus-death-toll-jumps-to-nearly-5-400-as-nursing-homes-included, last accessed May 1). We note the sensitivity to small errors of all epidemic models using exponential measures, such as the basic reproductive number, and the significant effects these minor errors have on the predicted outcome.

## CONCLUSIONS

Here, we present a model to estimate the effects of public interventions on the spread of COVID-19 that does not assume that interventions have identical results in different geographical and cultural settings. In contrast, our model uses *observational* data of mobility patterns in five environments to estimate changes in the transmission rate. Our model creates the possibility to track rapid changes in the spread, instantaneously and predict their consequences 3 weeks ahead in time. Therefore, our model enables governments to use anonymous real-time data to adjust their policies. We do foresee that such models will become incrementally more powerful as more detailed mobility data becomes available in the future.

## ACKNOWLEDGEMENTS

We acknowledge Claudio Bassot's contribution by sharing both the Imperial College London report and Google mobility data. Without this information, this study would not be possible. We are also grateful to various colleagues and friends that contributed to the discussion. Finally, we thank the authors of the Imperial College London report for making their data and model freely available.

### Funding

Financial support was provided by the Swedish Research Council for Natural Science, grant No. VR-2016-06301 and Swedish E-science Research Center. Computational

resources were provided by the Swedish National Infrastructure for Computing, grant No. SNIC-2019/3-319. The funders had no role in study design, data collection and analysis, decision to publish, or preparation of the manuscript.

### Grant Disclosures

The following grant information was disclosed by the authors:
Swedish Research Council for Natural Science: VR-2016-06301.
Swedish E-science Research Center.
Swedish National Infrastructure for Computing: SNIC-2019/3-319.

### Competing Interests

The authors declare that they have no competing interests.

### Author Contributions

- Patrick Bryant conceived and designed the experiments, performed the experiments, analysed the data, prepared figures and/or tables, authored or reviewed drafts of the paper, and approved the final draft.
- Arne Elofsson analysed the data, authored or reviewed drafts of the paper, and approved the final draft.

### Data Availability

Code and data are available at GitHub:
https://github.com/patrickbryant1/COVID19.github.io/tree/master/simulations/mobility.

### Supplemental Information

Supplemental information for this article can be found online at http://dx.doi.org/10.7717/peerj.9879#supplemental-information.

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
