# Peer review of "Estimating the impact of mobility patterns on COVID-19 infection rates in 11 European countries"

_PeerJ, doi:10.7717/peerj.9879_

## Round 0.1 · original submission · Major Revisions

Please carefully address all the issues pointed by the reviewers and revise your manuscript accordingly.

Reviewer 1 ·

Basic reporting

In the section of Introduction, Wuhan is city not province in China, please correct it.
The writing of the manuscript is professional and experimental design is reasonable and interesting. Interpretation of data supports the hypothesis and discussion of data provides significant value to future epidemiologists to conduct research on the epidemic of COVID19.

Experimental design

The experimental design is logical and provides valuable reference for future research on epidemics.

Validity of the findings

The current findings are quite interesting and consistent with the current strategies to curtail the COVID-19 pandemic disease.

Additional comments

The current research is quite interesting with logical design and preferable result interpretation.

·

Basic reporting

The paper suffers from many typos and inconsistencies in the formulae, which make the flow of the paper difficult to follow. Often, the impression is that the paper has been written too quickly. For example:

line 100: should use l_(k,t,m)
line 109: R_0 should be replaced why R_(0,m)
line 109: the Gaussian distribution also have a variance parameter. where is it?
line 130 and following: pi_m should depend on t as well
line 134: the construction that leads to this parameterization of the gamma distribution is not clear at all.
line 141:the cumulative number of deaths has not been defined so far. it comes afterwards, but in a different form
line 136: the logic underlying the discretisation step should be clarified

Experimental design

The model has the merit of introducing mobility data as predictors of the reproduction rate and, consequently, of the number of cases and number of deaths.

It suggests to do so using a Bayesian model, trying to extend the Imperial College London work which does not allows for a differential impact of mobility on R_0 among different countries.

The aim is valid but the methods employed to reach that aim are, in my opinion, weak. First of all, the logical flow of the model does not allow its interpretability. The authors should first explain the infection model, then the reproduction rate (depending on mobility) and finally the death model.

Instead, the reproduction rate model is introduced first, with very a strong assumption on the baseline R_0: that is assumed constant across countries and time. This assumptions imply no effects pf public health policies different from mobility restrictions (unrealistic); no effect of testing and health procedures, different in each country, and so on..

The infection model is then very cumbersome, starting with a Negative Binomial for the number of deaths (why not Poisson? it is not explained) , to a sequence of Gamma distributions which are convoluted in an unclear way, for the infection - to death part.
The number of cases follow a renewal process which depend on R_0 and on a gamma distribution for the incubation time which is , once more, constant in time and space: another unrealistic assumption.

In addition, all assumed distributions, and their parameterisations, are described without explanations, and without any intuition on why they are so.


Finally, the MCMC computations are described in a rather obscure way. For example: why eight chains were run? why to use only the Gelman and Rubin to detect convergence? as here the parameter distribution to be approximate is multivariate and not univariate.

Validity of the findings

The main finding (that grocery and pharmacy mobility explain 97% of the reduction of R_0) is very strong, and should be checked against a better, and clearer model, as described above.

Also the authors should compare their model with a SIR based model, that allows for the introduction of covariates, and allows for estimates to vary over time, as the health system of each country goes through different stress (and testing procedures) phases. See e.g. the Poisson autoregressive models developed by some authors to model covid-19 contagion (Agosto et al., 2020)

The paper has a robustness section, but it appears very limited, considered the amount of assumptions made by the authors.

Also, being the paper Bayesian, there is no attempt to show the posterior distribution of R_0 and the posterior distribution over the model space, which would give a better explanation of "the importance of each mobility data".

Additional comments

The paper deals with an important problem. The proposed methods should be better formulated, in a simpler and clearer way, also comparing the model not only with the Imperial College paper, but with the range of papers available to model covid-19 contagion.

To understand the proposal, the authors should pay much attention to the mathematical formulation, and specify each assumption and their rationale, as exemplified by my comments above

In particular, the authors should develop the model in the logical-statistical order of the variables: contagion model, R_0 model, contagion to death model.

The Bayesian approach should either be abandoned or stretched to a complete framework that includes the presentation of variable importance in terms of posterior probabilities of the different models (and features). Also convergence of the posterior distribution must be better checked

·

Basic reporting

The language is clear, and the author has also used enough references. The figures are also clear.

Experimental design

The author built a model to predict the epidemic to help the government making decisions based on the feedback of the model.
The authors are concerned that the COVID-19 will further spread upon the lifting of the restrictions around the world. Therefore, they are motivated to build the model to understand and predict.
They evaluated the problem of the current model and gave their modification to the model by addressing the effects of cultural and sociological differences.
They showed that 3-week forecast of their model has a smaller error for government to analyze if NPI are working.
Their model is trained from 30 days, which has enough dataset.
The only concern for this model is there are some assumptions that are made by the author, it would be great if the author can spend 1-2 sentence to explain the assumption.

Validity of the findings

They also validated the model using the 3-week data available and fit the data with the model to compare with ICL model.

---

## Round 0.2 · Minor Revisions

Please address the remaining issues pointed out by both reviewers and revise your manuscript.

·

Basic reporting

The paper is fine now

Experimental design

The paper is fine now

Validity of the findings

The paper is almost fine. The authors should include at least one reference on SIR models and their extension. I suggest Agosto, Giudici. (2020) A predictive model for COVID-19 dynamics. Risks. https://doi.org/10.3390/risks8030077

Additional comments

The authors took into account my suggestions in their revision. They should include at least one paper reference for Sir models as I suggested above

·

Basic reporting

The author sucessfully addressed the questions in the precious review. There are some minor points here:
PDF file, line 67-68, ‘Likely, different regions respond differently to the same NPIs, why it is vital to consider the effect of NPIs country wise’, this sentence does not read smooth.
Line 144-145: ‘The impact on R0 is assumed to be almost instantaneous, with an effect that decreases quickly with time, why a narrow gamma distribution was chosen’, this sentence does not read smooth

Line 230-231: ‘The R0 estimates are uncertain in the beginning of the epidemic when cases are few, why they are smoothed using one-week averages’ doesn’t read smooth

In table 2, last row, ‘Average absolute ror’ should be ‘Average absolute error’

Experimental design

The experiment design was reasonable.

Validity of the findings

The author validated the findings by using the model independently from the case data.

---

## Round 0.3 · accepted · Accept

Since all the remaining issues were adequately addressed, I am happy to accept your revised manuscript.